# Distinguishing Distributions
# When Samples Are Strategically Transformed

**Hanrui Zhang**
Duke University
Durham, NC 27708
hrzhang@cs.duke.edu

**Yu Cheng**
Duke University
Durham, NC 27708
yucheng@cs.duke.edu

**Vincent Conitzer**
Duke University
Durham, NC 27708
conitzer@cs.duke.edu

## Abstract

Often, a principal must make a decision based on data provided by an agent. Moreover, typically, that agent has an interest in the decision that is not perfectly aligned with that of the principal. Thus, the agent may have an incentive to select from or modify the samples he obtains before sending them to the principal. In other settings, the principal may not even be able to observe samples directly; instead, she must rely on signals that the agent is able to send based on the samples that he obtains, and he will choose these signals strategically.

In this paper, we give necessary and sufficient conditions for when the principal can distinguish between agents of "good" and "bad" types, when the type affects the distribution of samples that the agent has access to. We also study the computational complexity of checking these conditions. Finally, we study how many samples are needed.

## 1   Introduction

Anyone can have a bad day. Or a lucky one. Thus, in general, to determine with reasonable confidence who are the highly capable agents—whether they be people, companies, or anything else—we need to observe their output over an extended period of time. Moreover, capability is generally not one-dimensional, and who should be considered highly capable depends on what it is that we are looking for. Finally, the policy that we set to evaluate agents' output will in general affect how they strategically try to shape that output. Thus, we must choose our policy to enable the agents that are highly capable (according to our definition) to distinguish themselves from others.

**Example.** Suppose that there are researchers of different *types*. Specifically, suppose we have the following set of types:

$$\Theta = \{\text{TML-H}, \text{TML-L}, \text{AML-H}, \text{AML-L}\}$$

where "TML" stands for "theoretical machine learning," "AML" for "applied machine learning," and "L" and "H" for "low quality" and "high quality," respectively. Each researcher generates high-quality *ideas* (which we will in this paper refer to as *samples*) according to some probabilistic process. Suppose here the sample space is

$$S = \{\text{T}, \text{A}, \text{B}\}$$

where "T" stands for a purely theoretical idea without immediate applied significance, "A" for an applied idea without immediate theoretical significance, and "B" for an idea that has both theoretical and applied significance. Finally, suppose there are only 3 conferences: COLT, KDD, and NeurIPS (we will in this paper refer to papers published in these conferences as "signals").

$$\Sigma = \{\text{COLT}, \text{KDD}, \text{NeurIPS}\}$$

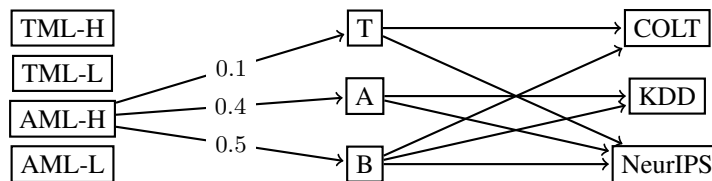

Figure 1: Illustration of the example.

A T or a B idea (sample) can be turned into a COLT paper (signal);[1] an A or a B idea can be turned into a KDD paper; and a T, A, or B idea can be turned into a NeurIPS paper.[2] Each idea, of course, can be published in only one conference.

Suppose a university would like to hire an AML-H researcher (but none of the other types). The faculty recruiting committee, unfortunately, is excessively lazy and only looks at the publication counts in the various venues. While the candidate researchers of course are committed to improving this terrible process once they get the job, for now their only concern is getting the job. In particular, everyone will attempt to pretend to be an AML-H researcher by sending their papers to the appropriate venues. But what exactly does this mean?

Suppose an AML-H researcher generates ideas at the following rates: 0.5 B, 0.4 A, 0.1 T. Moreover suppose that a TML-H researcher generates ideas at the following rates: 0.5 B, 0.1 A, 0.4 T. If the AML-H researcher sends all her papers to NeurIPS, then, even in the long run, she cannot distinguish herself from the TML-H researcher, who could do the same. On the other hand, if she sends strictly more than 0.6 of her ideas to KDD, then in the long run she will be able to distinguish herself from the TML-H researcher, because 0.4 of the latter's ideas cannot go to KDD.

Now consider the AML-L researcher. First, an easy case: suppose he generates ideas at the following rates: 0.4 B, 0.3 A, 0 T. (These numbers do not sum to 1, but this is not necessary, since they are rates. Equivalently, we can suppose him to have "the empty idea" $\varnothing$ with the remaining probability 0.3, which can be sent only to "the empty conference" where anything can be sent. This "empty signal" can also be used to model that the researchers sometimes only have ideas that they do not consider worth publishing, i.e., that they strategically select only a subset of their samples to pursue.) Clearly the AML-H researcher will in the long run distinguish herself from the AML-L researcher simply by the overall number of papers published (as long as the AML-H researcher does not unnecessarily send papers to the empty conference!). Alternatively, suppose the AML-L researcher generates ideas at the following rates: 0.4 B, 0.5 A, 0.1 T (so that the only weakness of the AML-L researcher relative to the AML-H researcher is that fewer of his ideas have both theoretical and applied significance). In this case, the AML-H researcher can, in the long run, distinguish herself from the AML-L researcher by sending strictly more than 0.5 of her ideas to COLT. Of course, this conflicts with what she needs to do distinguish herself from the TML-H researcher. Still, she can distinguish herself from both the TML-H and the AML-L researcher in the long run by, in odd-numbered years, sending strictly more than 0.6 of her ideas to KDD, and, in even-numbered years, sending strictly more than 0.5 of her ideas to COLT.

> In the long run we are all dead. —*John Maynard Keynes*

In reality, the candidates will have only finite time to prove themselves. Still, the lazy committee may hope to distinguish them with high probability. How many years suffice for this (and, therefore, should be the length of a typical Ph.D. program, potentially extended with a postdoctoral appointment)?

While this is example is a bit tongue-in-cheek, it is not hard to see that this basic phenomenon frequently occurs in society. People select from their opportunities and craft them to fit what they

think will appeal to future employers. A start-up company may select from its opportunities and craft them to fit what they think will impress future backers. In this paper, we introduce a general model that captures all these and other cases. Within this model, we characterize conditions under which agents of certain types can distinguish themselves from others, as well as how many samples are needed for this.

## 1.1 Related Work

Zhang et al. [19] study a related problem in which an agent draws samples and has to submit a subset of size $k$ of these samples to a principal, where $k$ is exogenous. In that paper, the motivation is that the principal can inspect only so many samples. In contrast, in this paper there is no such constraint, but samples can be modified or turned into signals according to a given (arbitrary) graph. This paper also allows for uncertainty about how many samples an agent has available, via the "empty sample/signal" trick illustrated in the introductory example.

Our setting is related to mechanism design with *partial verification* [8, 18], where an agent's type restricts which signals he can send. This can be thought of as corresponding to the special case of our model in which an agent only has a single sample which is fully determined by his type. More generally, our setting is related to the literature in economics on *signaling* (along the lines of [16]). However, our model does not involve the agents taking any costly actions. There is other work that generalizes the partial verification setting to allow costly signaling [12, 13], motivated in part by *strategic classification* settings where agents are being classified but they can strategically change their features at some cost (as also studied in [9]).[3] In contrast to this line of work, in this paper we consider settings where a single agent with a single type *repeatedly* generates samples according to a distribution (which are then strategically transformed into signals). This allows us to study the question of how many samples are needed to, with high confidence, distinguish types from each other.

Our results can be viewed as generalizations of classical results in *efficient statistics*, and in particular, results for learning and testing discrete distributions, to strategic settings. One of our main results, Theorem 6, relies on a subroutine which generalizes the folklore result of estimating discrete distributions. Another main result, Theorem 7, uses as a building block the sample-optimal identity testing algorithm for discrete distributions [6, 17]. Theorem 7 generalizes their algorithm into an environment where samples can be strategically modified according to a partial order.

## 2 Preliminaries

For a set $S$, we use $\Delta(S)$ to denote the set of probability distributions over $S$. Given a distribution $x \in \Delta(S)$, we use $x(i)$ to denote the probability mass on the element $i \in S$, and $x(A)$ to denote the total probability mass on the set $A \subseteq S$. We are generally interested in distinguishing one or more *good* distributions from one or more *bad* distributions (where good and bad are determined by what we are looking for). We use $g$ to denote the good distribution, and $b$ to denote the bad distribution. (We use $(g_i)_i$ and $(b_i)_i$ when there are multiple good/bad distributions.) The agent, depending on his type being either good or bad, draws $n$ samples i.i.d. from either $g$ or $b$. How samples can be turned into signals is represented by a bipartite graph $G = (S \cup \Sigma, E)$ between the (discrete) sample space $S$ and the (discrete) signal space $\Sigma$. An agent must convert each sample into a signal and then submit all $n$ signals to the principal. $E$ specifies which signals are valid for each sample: a sample $s \in S$ can be converted into a signal $\sigma \in \Sigma$ iff $(s, \sigma) \in E$.

Note that our model generalizes each of the following models:

1. The agent can choose to omit samples. We can add an "empty signal" to $\Sigma$, where converting a sample $s$ to the empty signal corresponds to not reporting $s$.
2. The agent may or may not receive a sample in each round. E.g., in the example where samples correspond to ideas and signals correspond to papers, in some rounds the agent may not have any (worthwhile) idea. We can add an "empty sample" in $S$ which can only be converted to the empty signal.

3. The signal space is the same as the sample space: $S = \Sigma$. In this case it is more natural to replace the bipartite graph by one that has only one copy of each sample/signal, is no longer bipartite, and that represents the possibility of changing sample/signal $u$ to sample/signal $v$ by a directed edge $(u, v)$.

We will be interested in the probability of accepting good or bad types after $T$ rounds (i.e., after the agent draws $T$ samples). We call the $T$ signals submitted a *report* $\mathcal{R} \in \Sigma^T$. The principal gets to choose an acceptance function (or policy, which could be randomized) $f : \mathcal{R} \to \{0, 1\}$ that maps the report into a binary decision. Her goal is to accept the good agent and reject the bad agent. The agent wants to be accepted regardless of his type. The principal can thus make two types of mistakes: false-positive (or type 1 error) when she accepts a bad agent, and false-negative (type 2 error) when she rejects a good agent. The principal wants to minimize the maximum probability of making either type of mistakes.

We recall the following definition of the total variation distance:

**Definition 1** (Total Variation Distance). The total variation distance between two distributions $x, y \in \Delta(\Sigma)$ over support $\Sigma$ is defined to be

$$d_{\mathrm{TV}}(x, y) = \frac{1}{2}\|x - y\|_1 = \frac{1}{2}\sum_{\sigma \in \Sigma}|x(\sigma) - y(\sigma)| = \max_{A \subseteq \Sigma}(x(A) - y(A)).$$

In our setting, the total variation distance provides a good way to measure the closeness between two *signal* distributions, which are observable by the principal. We will generalize this definition to our strategic setting, to measure how close two distributions over the *sample* space are to each other.

## 3 Basic Structural Results

In this section, we define a notion that we term "directed total variation distance" $d_{\mathrm{DTV}}$. For two distributions $x$ and $y$ over samples, $d_{\mathrm{DTV}}(x, y)$ measures how well $x$ can distinguish itself from $y$ in our strategic setting. As we will see in the later sections, $d_{\mathrm{DTV}}$ is a central notion in this paper, and often dictates the number of samples we need to distinguish the two distributions under strategic reporting.

We first give the formal definitions of *reporting strategies* and the directed total variation distance $d_{\mathrm{DTV}}(x, y)$. Then we define another notion $\mathrm{MaxSep}(x, y)$ that measures how well $x$ can distinguish itself from $y$ from the principal's perspective, using separating sets instead of reporting strategies. Given these definitions, we present one of our key structural results (Proposition 1), which shows that the two notions are equivalent.

Before investigating distinguishing distributions under strategic reporting, we first generalize the classical measure of how close two distributions are, $d_{\mathrm{TV}}$, to our strategic setting. We first give a formal definition the *reporting strategy* used by the agents.

**Definition 2** ((Single-Round) Reporting Strategy). Given $x \in \Delta(S), \alpha \in \Delta(\Sigma)$, we say $x$ *can report* $\alpha$ $(x \to \alpha)$, if there exist a *reporting strategy* $R = \{r_{s,\sigma}\}_{(s,\sigma) \in E}$ satisfying:

- $r_{s,\sigma} \geq 0$ for all $(s, \sigma) \in E$.
- For each $s \in S$, $\sum_{\sigma:(s,\sigma) \in E} r_{s,\sigma} = 1$.
- For each $\sigma \in \Sigma$, $\sum_{s:(s,\sigma) \in E} x(s) \cdot r_{s,\sigma} = \alpha(\sigma)$.

We say $x$ reports $\alpha$ by strategy $R$ $(x \to_R \alpha)$.

In other words, when each sample $s \in S$ is drawn from the distribution $x$ and given this sample the agent is reporting $\sigma \in \Sigma$ with probability $r_{s,\sigma}$, the resulting distribution over the signal space is exactly $\alpha$. For a fixed sample or a random variable $s$, we use $R(s) \in \Delta(\Sigma)$ to denote the random variable whose distribution over the signal space is induced by $\{R_{s,\sigma}\}_{\sigma \in \Sigma}$.

Given the definition of reporting strategies, we are ready to generalize $d_{\mathrm{TV}}$ to our setting. Intuitively, $x$ chooses a report first, and then $y$ chooses a report in response; they play a zero-sum game where $x$ wants the reports to be as far away from each other as possible. $d_{\mathrm{DTV}}(x, y)$ is the value of this two-player game when $x$ must choose a report (i.e., a pure strategy) first, which measures how far $x$ can stay away from $y$.

**Definition 3** (Directed Total Variation Distance). Given $(S, \Sigma, E)$, the directed total variation distance between two distributions $x, y \in \Delta(S)$ over the sample space $S$ is defined to be

$$d_{\mathrm{DTV}}(x, y) = \max_{\alpha: x \to \alpha} \min_{\beta: y \to \beta} d_{\mathrm{TV}}(\alpha, \beta).$$

Directed total variance distance nicely characterizes the distance between two distributions from the agent's perspective, but it is not immediately clear how that might help the principal. In particular, are two distributions easily separable by setting an appropriate policy if they have large directed total variation distance? To study this, we introduce several concepts to model the problem from the principal's perspective.

**Definition 4** (Preimage of Signals). For any set of signals $A \subseteq \Sigma$, the preimage $\mathrm{pre}(A)$ of $A$ is defined to be the set of samples which can be mapped to a signal in $A$. That is

$$\mathrm{pre}(A) = \{s \in S \mid \exists \sigma \in A, \text{ s.t. } (s, \sigma) \in E\}.$$

The principal could label a set $A$ of signals as "good" signals and simply measure how many good signals the agent is able to send. Ideally, this $A$ is chosen so that a good agent can send (significantly) more signals in $A$ than a bad agent. This inspires the following definitions.

**Definition 5** (Separation). For any $A \subseteq \Sigma$, if $x(\mathrm{pre}(A)) - y(\mathrm{pre}(A)) = \epsilon > 0$, then we say $A$ separates $x$ from $y$ by a margin of $\epsilon$.

**Definition 6** (Max Separation). The max separation of $x \in \Delta(S)$ from $y \in \Delta(S)$ over the sample space $S$ is defined to be $\mathrm{MaxSep}(x, y) = \max_{A \subseteq \Sigma}(x(\mathrm{pre}(A)) - y(\mathrm{pre}(A)))$.

We now draw the connection between the agent's and the principal's perspectives. The following proposition can be viewed as a generalization of the classic Hall's Marriage Theorem. Proposition 1 states that $g$ can distinguish itself from $b$ under strategic reporting iff there exists a subset $A^*$ of signals so that $g$ can generate more signals in $A^*$ than $b$. Equivalently, the best reporting strategy for $g$ is to focus on a subset $A^*$ of the signal space, and try to convert samples into signals in $A^*$ whenever possible.

**Proposition 1.** *For any $x, y \in \Delta(S)$, $d_{\mathrm{DTV}}(x, y) = \mathrm{MaxSep}(x, y)$.*

The proof of the proposition, as well as all other proofs, is deferred to the appendix. This equivalence between $d_{\mathrm{DTV}}$ and $\mathrm{MaxSep}$ not only is a nice structural result; Proposition 1 plays a substantial part in our main algorithmic results.

It is worth noting that $d_{\mathrm{DTV}}(x, y)$ in general is not equal to $d_{\mathrm{DTV}}(y, x)$. However, the triangle inequality still holds for $d_{\mathrm{DTV}}$, which also enables some of our main results.

**Proposition 2.** *For any $x, y, z \in \Delta(S)$, $d_{\mathrm{DTV}}(x, y) + d_{\mathrm{DTV}}(y, z) \geq d_{\mathrm{DTV}}(x, z)$.*

## 4 Structural and Computational Results in the General Case

In this section, we define adaptive and non-adaptive reporting strategies (Definition 7), and the accepting probabilities of the optimal reporting strategies after $T$ rounds (Definition 8). At a high level, we give a tight characterization result on when there exists a policy that can distinguish $g$ from $b$ under strategic reporting, and provide an asymptotically tight bound on the sample complexity of the optimal policy. Moreover, we show that while our structural result is clean and tight, it is computationally hard to check if the condition holds. That is, in the general case, it is NP-hard to determine whether there is a policy that can distinguish $g$ from $b$.

More specifically, we first show that there exists a policy that can distinguish $g$ from $b$ in the limit (when $T \to \infty$) iff $d_{\mathrm{DTV}}(g, b) > 0$ (Theorem 1). Next, we give an asymptotically tight sample complexity bound of $T = \Theta(1/\epsilon^2)$ when $d_{\mathrm{DTV}}(g, b) = \epsilon$ and we want to distinguish $g$ from $b$ with high constant probability (Theorem 3). We then extend the existence result to more general settings when there are multiple good and bad distributions (Theorem 4). Finally, we show that it is NP-hard to decide if we are in the case where $d_{\mathrm{DTV}}(g, b) = 0$ or $d_{\mathrm{DTV}}(g, b) > \frac{1}{\mathrm{poly}(m,n)}$ (Theorem 2).

We start with the definition of adaptive reporting strategies.

**Definition 7** (Adaptive Reporting Strategy). An adaptive reporting strategy $\mathcal{R} = (R^1, \ldots, R^T)$ is a sequence of (different) reporting strategies. The signal $\sigma^i$ at time $i$ is obtained by applying $R^i$ to

the sample $s^i$ at time $i$. $R^i = R^i(\sigma^1, \ldots, \sigma^{i-1})$ may depend on all past signals. A reporting strategy is non-adaptive if $R^i = R^1$ for any $i$ and $(\sigma^1, \ldots, \sigma^{i-1})$, and adaptive otherwise. For an adaptive policy $\mathcal{R} = (R^1, \ldots, R^T)$, we interchangeably write $\sigma^i = R^i(s^i \mid \sigma^1, \ldots, \sigma^{i-1})$ to indicate the dependence of $R^i$ on $\sigma^1, \ldots, \sigma^{i-1}$.

When we analyze the quality of a fixed $T$-round policy $f$, we are interested in the probability that $f$ accepts $g$ or $b$ after $T$ rounds, when the agent (of either type) best-responds to $f$.

**Definition 8** (Acceptance Probabilities of the Best Reporting Strategies). Given $x \in \Delta(S)$, $T \in \mathbb{N}$, and the principal's policy $f$, let the acceptance rate under adaptive / non-adaptive reporting respectively be

$$p_{\mathrm{ada}}(f, x, T) = \max_{\mathcal{R}=(R^1,\ldots,R^T)} \mathbb{E}[f((R^i(s^i))_{i\in[T]})],$$

$$p_{\mathrm{non}}(f, x, T) = \max_{\mathcal{R}=(R,\ldots,R)} \mathbb{E}[f((R^i(s^i))_{i\in[T]})]$$

where the expectations are taken over $T$ i.i.d. samples $(s^i)_i$ drawn from $x$. Observe that $p_{\mathrm{ada}}(f, x, T) \geq p_{\mathrm{non}}(f, x, T)$ for any $f$, $x$ and $T$.

Intuitively, if $d_{\mathrm{DTV}}(g, b) = 0$, then the bad distribution can mimic the good distribution perfectly in the signal space, no matter what reporting strategy $g$ uses. Therefore, it is impossible to distinguish $g$ from $b$. The next theorem formalizes this intuition. In particular, even if $g$ reports adaptively, $b$ can still mimic $g$'s conditional reporting strategy in every situation (i.e., for every combination of previously reported signals).

**Theorem 1** (Separability in the Limit). *Given good and bad distributions $g$ and $b$:*

*(i) If $d_{\mathrm{DTV}}(g, b) > 0$, then there exists a policy $f$ such that*

$$\lim_{T\to\infty} (p_{\mathrm{non}}(f, g, T) - p_{\mathrm{ada}}(f, b, T)) = 1.$$

*That is, $f$ accepts $g$ and rejects $b$ with probability $1$ in the limit.*
*(ii) If $d_{\mathrm{DTV}}(g, b) = 0$, then for any policy $f$ and any $T$,*

$$p_{\mathrm{ada}}(f, g, T) \leq p_{\mathrm{ada}}(f, b, T), p_{\mathrm{non}}(f, g, T) \leq p_{\mathrm{non}}(f, b, T).$$

*That is, no policy can separate $g$ from $b$, regardless of whether the setting is adaptive.*

The next theorem states that while our characterization result (Theorem 1) is clean and tight (we can distinguish iff $d_{\mathrm{DTV}}(g, b) > 0$), it is in fact computationally hard to check if this condition holds. Intuitively, Theorem 2 constructs an instance where the good distribution needs to focus on as few signals as possible. The parameters are chosen carefully so that it is crucial that $g$ finds a subset of signals $A \subseteq \Sigma$ with minimum cardinality that covers the support of $g$.

**Theorem 2** (hardness of checking separability). *Given $x, y \in \Delta(S)$, it is NP-hard to distinguish between the following two cases: (1) $d_{\mathrm{DTV}}(x, y) = 0$ and (2) $d_{\mathrm{DTV}}(x, y) \geq \frac{1}{\mathrm{poly}(m,n)}$, or equivalently, to determine the existence of a set $A \subseteq \Sigma$ such that $x(\mathrm{pre}(A)) - y(\mathrm{pre}(A)) \geq \frac{1}{\mathrm{poly}(m,n)}$.*

Note that the hardness of checking the existence of separating sets implies the hardness of finding any separating set given that $d_{\mathrm{DTV}}(x, y) > 0$. This is because given an algorithm for the latter problem, one could run that algorithm without knowing whether $d_{\mathrm{DTV}}(x, y) > 0$ and see if it succeeds. Either the algorithm returns a separating set, or we know it must be the case that $d_{\mathrm{DTV}}(x, y) = 0$ and no separating set exists.

Next, we focus on the case when there are finitely many samples. Theorem 3 is more refined than Theorem 1, in that it gives a tight sample complexity bound instead of only talking about distinguishing $g$ and $b$ in the limit.

**Theorem 3** (Sample Complexity with Two Distributions). *For any $g$ and $b$ such that $d_{\mathrm{DTV}}(g, b) \geq \epsilon$:*

- *There is a policy $f$ such that for any $\delta > 0$ and $T \geq 2\ln(1/\delta)/\epsilon^2$, $p_{\mathrm{non}}(f, g, T) \geq 1 - \delta$ and $p_{\mathrm{ada}}(f, b, T) \leq \delta$.*
- *When $d_{\mathrm{DTV}}(g, b) = \epsilon$ and $T = o(1/\epsilon^2)$, for any $f$, $p_{\mathrm{non}}(f, g, T) - p_{\mathrm{non}}(f, b, T) < \frac{1}{3}$.*

Theorem 3 can be generalized to the case where there are multiple good and bad distributions. First, suppose there is one good distribution and multiple bad distributions. As long as $d_{\mathrm{DTV}}(g, b_j) \geq \epsilon$ for every bad distribution $b_j$, we can use the testing algorithm in Theorem 3 to distinguish them in $T = O(1/\epsilon^2)$ rounds (with high constant probability). We potentially need to do so separately for every bad distribution, paying an extra factor of $\Omega(\ell)$ in the sample complexity if there are $\ell$ bad distributions. If there are $k$ good distributions, then we can run the $k$ testers in parallel, paying an additional factor of $\log(k)$ in the sample complexity to boost the success probability so that we can take a union bound.

**Theorem 4** (Multiple Good and Bad Distributions, the General Case). *For any $g_1, \ldots, g_k$ and $b_1, \ldots, b_\ell$ such that $d_{\mathrm{DTV}}(g_i, b_j) \geq \epsilon$ for any $i \in [k]$ and $j \in [\ell]$, there is a policy $f$ such that: For any $\delta > 0$ and $T \geq 2\ell \ln(k\ell/\delta)/\epsilon^2$, $p_{\mathrm{ada}}(f, g_i, T) \geq 1 - \delta$ for any $i \in [k]$, and $p_{\mathrm{ada}}(f, b_j, T) \leq \delta$ for any $j \in [\ell]$.*

We note that the policy in Theorem 4 requires the good distribution to report in different ways, which is not possible with a non-adaptive strategy according to our definition. In particular, the good distribution must know which bad distribution it is up against in each phase, and report accordingly. As our introductory example shows, this is in fact necessary when there are multiple bad distributions.

# 5 When Signals Are Partially Ordered

In many real-world situations, the sample and signal spaces are structured. For example, when a band is recruiting new members, applicants may be asked to submit video recordings of themselves playing. An applicant would probably videotape herself playing for an entire event as a sample, and then crop the recording to create a signal that demonstrates only her best performance. This cropping procedure is irreversible: the complete recording may be cropped to keep a part, but from a part, it is impossible to recover the full recording. The signal space in this scenario is partially ordered by the cropping procedure—the samples/signals can be transformed in one direction (shortening), but never the other. Also, there is a "default" signal for each sample, which is simply to submit the complete recording without cropping. The default signal can be transformed into any signal that can be reported from this sample. In this section, we consider the following abstraction of such scenarios:

- $S = \Sigma$,
- $(s, s) \in E$ for any $s \in S$,
- $(s, t) \in E$ and $(t, u) \in E \implies (s, u) \in E$, and
- $E$ is acyclic except for self-cycles.

This abstraction also covers, for example, scenarios where the agent can choose to hide certain samples—any sample can be transformed into a non-sample, but not reversely. Note that given the above conditions, the sample/signal space is essentially a partially ordered set, where a sample can only be transformed according to this partial order. Let $n = |S|$ be the cardinality of the sample/signal space.

We first show some useful structural results in the partially ordered case. The following proposition demonstrates that the revelation principle holds in this case.

**Proposition 3** (Revelation Principle). *For any policy $f$:*

- *There exists a policy $f'$ such that for any $x \in \Delta(S)$, $T \in \mathbb{N}$,*

$$p_{\mathrm{non}}(f, x, T) = p_{\mathrm{non}}(f', x, T) = \mathbb{E}[f'((s^i)_i)].$$

- *There exists a policy $f''$ such that for any $x \in \Delta(S)$, $T \in \mathbb{N}$,*

$$p_{\mathrm{ada}}(f, x, T) = p_{\mathrm{ada}}(f'', x, T) = p_{\mathrm{non}}(f'', x, T) = \mathbb{E}[f''((s^i)_i)].$$

In other, non-learning contexts in mechanism design, whether the revelation principle holds is often an aspect that determines whether the computational problems therein are tractable. We will see that this is also the case for our problem—the revelation principle enables efficient computation of the max separation, and therefore efficient policies in a quite natural way.

The next proposition simplifies the definition of $d_{\mathrm{DTV}}$ in the partially ordered case, based on the insight that, per the revelation principle, the best way for $x$ to avoid being mimicked by $y$ is to always report the unmodified samples.

**Proposition 4** ($d_{\mathrm{DTV}}$ Simplified). *In the transitive case, $d_{\mathrm{DTV}}(x, y) = \min_{y \to y'} d_{\mathrm{TV}}(x, y')$.*

This also gives us an efficient algorithm for finding the set that supports the max separation $\mathrm{MaxSep}(x, y)$ of $x$ from $y$:

**Corollary 1** (Efficient Computation of Max Separation). *Given any $x, y \in \Delta(S)$, there is a poly-time algorithm which computes a set $A^*$ satisfying $x(\mathrm{pre}(A^*)) - y(\mathrm{pre}(A^*)) = \mathrm{MaxSep}(x, y)$.*

We show in Theorem 5 that in the partially ordered case we can separate multiple good distributions from multiple bad ones with much smaller overhead. The proof of Theorem 5 is similar to that of Theorem 4. The only difference is that, because of the revelation principle, we no longer require good distributions to report adaptively.

**Theorem 5** (Multiple Good and Bad Distributions: The Partially Ordered Case). *For any $g_1, \ldots, g_k$ and $b_1, \ldots, b_\ell$ where $d_{\mathrm{DTV}}(g_i, b_j) \geq \epsilon$ for any $i \in [k]$, $j \in [\ell]$, there is a policy $f$ such that: For any $\delta > 0$ and $T \geq 2 \ln(k\ell/\delta)/\epsilon^2$, $p_{\mathrm{non}}(f, g_i, T) \geq 1 - \delta$ for any $i \in [k]$, and $p_{\mathrm{ada}}(f, b_j, T) \leq \delta$ for any $j \in [\ell]$.*

In the partially ordered case, we cannot only deal with multiple good and bad distributions much more efficiently, but also deal with any bad distribution using a single sample-efficient policy. Before stating the result, recall the following definition of the *width* of a partially ordered set.

**Definition 9** (Width of Partially Ordered Sets). The width $\rho(G)$ of a partially ordered set represented as graph $G = (S, E)$ is defined to be $\rho(G) = \max\{|A| \mid A \subseteq S, \forall s_1, s_2 \in A, (s_1, s_2) \notin E\}$. In other words, the width is the maximum size of a set $A \subseteq S$ where any two elements in $A$ are not comparable. Such a set $A$ is called an *anti-chain*.

We now provide our generic policy, whose sample complexity, quite surprisingly, depends roughly linearly on the width of the sample space.

**Theorem 6** (Efficient Policy against Any Bad Distribution). *For any $g \in \Delta(S)$, there is a policy $f$ such that for any $\delta > 0$, and $T \geq \frac{2\rho \ln(1 + n/\rho) \ln(1/\delta)}{\epsilon^2}$: (1) $p_{\mathrm{non}}(f, g, T) \geq 1 - \delta$, and (2) for any $b$ such that $d_{\mathrm{DTV}}(g, b) \geq \epsilon$, $p_{\mathrm{ada}}(f, b, T) \leq \delta$. Moreover, the outcome of the policy can be computed in polynomial time.*

The above policy is able to detect any bad distribution with adaptive reporting. For bad distributions without adaptive reporting, when $\rho = \Omega(\sqrt{n}/\log n)$, the following policy achieves even better sample complexity.

**Theorem 7** (Efficient Policy against Non-adaptive Bad Distributions). *For any $g \in \Delta(S)$, there is a policy $f$ such that for any $\delta > 0$, with $T = O\left(\frac{\sqrt{n}\ln(1/\delta)}{\epsilon^2}\right)$ samples: (1) $p_{\mathrm{non}}(f, g, T) \geq 1 - \delta$, and (2) for any $b$ such that $d_{\mathrm{DTV}}(g, b) \geq \epsilon$, $p_{\mathrm{non}}(f, b, T) \leq \delta$. Moreover, the outcome of the policy can be computed in polynomial time.*

## 6 Future research

In this paper, we have focused on distinguishing good and bad types with near certainty. In reality, the number of available samples may not always be sufficient for this. If so, it may be worthwhile to move beyond simple acceptance and rejection decisions to a more general mechanism design setup. For example, when the signals we receive from an agent are not decisive one way or another, perhaps an intermediate outcome between rejection and acceptance allows us to improve our objective, by avoiding the damage of either accepting a bad type or rejecting a good type. One may also consider settings in which signaling is costly (or at least sending high-quality signals comes at an effort cost, in line with traditional signaling models [16]) or in which agents can in fact improve their actual types via some investment cost. Any of these directions would further enrich the specific connections between mechanism design and learning theory that we have begun to explore in this paper (and that in turn complement other fascinating connections between these topics that have earlier been established by others [1, 11, 2, 10, 3, 4, 14, 7]).

**Acknowledgements.** We are thankful for support from NSF under awards IIS-1814056 and IIS-1527434. We also thank anonymous reviewers for helpful comments.

## Footnotes

[1]Of course, having the basic idea is generally only a small part of the work that needs to be done for a conference paper; but for our purposes here, we may imagine that the idea incorporates all the work that needs to be done.

[2]We use the names of actual conferences strictly for amusement value, and while we think our example roughly aligns with the focus of these conferences, we do not mean to imply anything about their selectivity (all these ideas are high-quality) or open-mindedness. We also do not mean to imply anything about other conferences—e.g., ICML could just as well have been used instead of NeurIPS—or (in what follows) about different types of researchers or the priorities and effort levels of actual hiring committees.

[3]Other work that models strategic agents manipulating the data that they submit [5, 15] concerns *aggregating* the data of multiple agents into a *single* outcome that all these agents care about; as such, this is less related to our model here, as here we are interested in determining a given single agent's type rather than choosing a single outcome that affects multiple agents.

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
