[Supplementary Material]

# Distinguishing Distributions
# When Samples Are Strategically Transformed

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

 # A Omitted Proofs From Section 3

388 We need the following fact:

389 **Proposition 5** (Saturation). *If $x \to \alpha$, then for any $A \subseteq \Sigma$,*

$$x(\mathrm{pre}(A)) \geq \alpha(A).$$

390 *Moreover, there exists $\alpha_A$ where $x \to \alpha_A$, such that*

$$x(\mathrm{pre}(A)) = \alpha_A(A).$$

391 *We call the corresponding reporting strategy that achieves $x \to \alpha_A$ "saturating" for A.*

392 *Proof of Proposition 5.* Let $R = \{r_{s,\sigma}\}_{(s,\sigma)\in E}$ be the reporting strategy by which $x$ reports $\alpha$.

$$
\begin{aligned}
x(\mathrm{pre}(A)) &= \sum_{s\in\mathrm{pre}(A))} x(s) \\
&\geq \sum_{s\in\mathrm{pre}(A)}\sum_{\sigma\in A} r_{s,\sigma}x(s) && (\textstyle\sum_{\sigma\in A} r_{s,\sigma} \leq 1) \\
&= \sum_{\sigma\in A}\sum_{s:(s,\sigma)\in E} r_{s,\sigma}x(s) \\
&= \sum_{\sigma\in A}\alpha(\sigma) && (\text{definition of } R) \\
&= \alpha(A).
\end{aligned}
$$

393 Now we show $\alpha_A$ exists by constructing the corresponding reporting strategy. Let $R' = \{r'_{s,\sigma}\}$ be
394 any reporting strategy satisfying: if $s \in \mathrm{pre}(A)$, $r'_{s,\sigma} = 0$ for all $\sigma \notin A$. Such an $R'$ exists because
395 by the definition of $\mathrm{pre}(A)$, for every $s \in \mathrm{pre}(A)$, there is at least one $\sigma \in A$ that connects to $s$.

396 Now for any $s \in \mathrm{pre}(A)$,

$$\sum_{\sigma\in A} r'_{s,\sigma} = 1.$$

397 Hence, for this reporting strategy, the single inequality in the derivation above becomes an equality,
398 allowing us to conclude $x(\mathrm{pre}(A)) = \alpha_A(A)$. □

399 *Proof of Proposition 1.* We first show $\mathrm{MaxSep}(x,y) \leq d_{\mathrm{DTV}}(x,y)$. Let $A^* =$
400 $\mathrm{argmax}_A(x(\mathrm{pre}(A)) - y(\mathrm{pre}(A)))$.

$$
\begin{aligned}
d_{\mathrm{DTV}}(x,y) &= \max_{\alpha:x\to\alpha}\min_{\beta:y\to\beta} d_{\mathrm{TV}}(\alpha,\beta) \\
&\geq \max_{\alpha:x\to\alpha}\min_{\beta:y\to\beta} \sum_{\sigma\in A^*}\max\{\alpha(\sigma)-\beta(\sigma),0\} && (\text{Definition 1 of } d_{TV}) \\
&\geq \max_{\alpha:x\to\alpha}\min_{\beta:y\to\beta} (\alpha(A^*)-\beta(A^*)) \\
&\geq \max_{\alpha:x\to\alpha} (\alpha(A^*)-y(\mathrm{pre}(A^*))) && (\text{Proposition 5}) \\
&= x(\mathrm{pre}(A^*))-y(\mathrm{pre}(A^*)) && (\text{Proposition 5, existence of saturating distribution}) \\
&= \mathrm{MaxSep}(x,y).
\end{aligned}
$$

401 Now we show $\mathrm{MaxSep}(x,y) \geq d_{\mathrm{DTV}}(x,y)$. Let $\alpha^*$ be a signal distribution reported by $x$ that
402 achieves $d_{\mathrm{DTV}}(x,y)$. Let $\beta^*$ be a signal distribution reported by $y$ that best-response to $\alpha^*$, where
403 we require as a tie-breaker that $\beta^*$ minimizes the number of signals $\sigma$ with $\alpha^*(\sigma) \geq \beta^*(\sigma)$.

404 Let $A^* = \{\sigma \mid \alpha^*(\sigma) \geq \beta^*(\sigma)\}$. We will show that $A^*$ separates $x$ from $y$ by a margin of
405 $d_{\mathrm{DTV}}(x,y)$.

406 We first show that $\beta^*(A^*) = y(\mathrm{pre}(A^*))$. Suppose otherwise $\beta^*(A^*) < y(\mathrm{pre}(A^*))$. Let $R =$
407 $\{r_{s,\sigma}\}$ be the reporting strategy that gives $y \to \beta^*$. We know that there exists some $s_0 \in \mathrm{pre}(A^*)$

with $y(s_0) > 0$ where $R$ does not convert all probability mass on $s_0$ into signals in $A^*$. Formally, we have $\sum_{\sigma \in A^*:(s_0,\sigma)\in E} r_{s_0,\sigma} < 1$. Consider any $\sigma_1, \sigma_2 \in \Sigma$ satisfying: $\sigma_1 \notin A^*$, $(s_0,\sigma_1) \in E$, $r_{s_0,\sigma_1} > 0$, $\sigma_2 \in A^*$, and $(s_0,\sigma_2) \in E$. We have $\alpha^*(\sigma_1) < \beta^*(\sigma_1)$ and $\alpha^*(\sigma_2) \geq \beta^*(\sigma_2)$. Now we discuss the following two cases and show there is a contradiction in both cases.

- If $\alpha^*(\sigma_2) > \beta^*(\sigma_2)$, then by moving

$$\min\{r_{s_0,\sigma_1}y(s_0), \beta^*(\sigma_1) - \alpha^*(\sigma_1), \alpha^*(\sigma_2) - \beta^*(\sigma_2)\} > 0$$

  mass from $\sigma_1$ to $\sigma_2$, $y$ can report $\beta'$ such that $d_{\mathrm{TV}}(\alpha^*, \beta') < d_{\mathrm{TV}}(\alpha^*, \beta^*)$, a contradiction.
- If $\alpha^*(\sigma_2) = \beta^*(\sigma_2)$, then by moving

$$\min\{r_{s_0,\sigma_1}y(s_0), (\beta^*(\sigma_1) - \alpha^*(\sigma_1))/2\} > 0$$

  mass from $\sigma_1$ to $\sigma_2$, $y$ can report $\beta'$ such that $d_{\mathrm{TV}}(\alpha^*, \beta^*) = d_{\mathrm{TV}}(\alpha^*, \beta')$. But now $\alpha^*(\sigma_2) - \beta'(\sigma_2) < 0$, and for any $\sigma \neq \sigma_2$, the sign of $\alpha^*(\sigma) - \beta'(\sigma)$ is the same as that of $\alpha^*(\sigma) - \beta^*(\sigma)$. So we have

$$|\{\sigma \mid \alpha^*(\sigma) \geq \beta^*(\sigma)\}| > |\{\sigma \mid \alpha^*(\sigma) \geq \beta'(\sigma)\}|,$$

  which contradicts the choice of $\beta^*$.

Now given that $y(\mathrm{pre}(A^*)) = \beta^*(A^*)$, we have

$$
\begin{aligned}
\mathrm{MaxSep}(x,y) &= \max_A(x(\mathrm{pre}(A)) - y(\mathrm{pre}(A))) \\
&\geq x(\mathrm{pre}(A^*)) - y(\mathrm{pre}(A^*)) \\
&\geq \alpha^*(A^*) - y(\mathrm{pre}(A^*)) \qquad\qquad \text{(Proposition 5)} \\
&= \alpha^*(A^*) - \beta^*(A^*) \\
&= d_{\mathrm{TV}}(\alpha, \beta) \\
&= d_{\mathrm{DTV}}(x, y). \qquad\qquad\qquad\qquad\qquad \square
\end{aligned}
$$

*Proof of Proposition 2.* Let $A^* = \mathrm{argmax}_A(x(\mathrm{pre}(A)) - z(\mathrm{pre}(A)))$. We have

$$
\begin{aligned}
d_{\mathrm{DTV}}(x,y) + d_{\mathrm{DTV}}(y,z) &= \mathrm{MaxSep}(x,y) + \mathrm{MaxSep}(y,z) \\
&= \max_A(x(\mathrm{pre}(A)) - y(\mathrm{pre}(A))) + \max_A(y(\mathrm{pre}(A)) - z(\mathrm{pre}(A))) \\
&\geq (x(\mathrm{pre}(A^*)) - y(\mathrm{pre}(A^*))) + (y(\mathrm{pre}(A^*)) - z(\mathrm{pre}(A^*))) \\
&= x(\mathrm{pre}(A^*)) - z(\mathrm{pre}(A^*)) \\
&= \mathrm{MaxSep}(x,z) \\
&= d_{\mathrm{DTV}}(x,z). \qquad\qquad\qquad\qquad\qquad\qquad\qquad\qquad\quad \square
\end{aligned}
$$

# B  Omitted Proofs From Section 4

*Proof of Theorem 1.* Part (i) follows from Theorem 3.

For part (ii), suppose $d_{\mathrm{DTV}}(g, b) = 0$. Let $s_g^i$ (resp. $s_b$) be a random variable that denotes the sample drawn from $g$ (resp. $b$) at time $i$. Abusing notation, for two random variables $X$ and $Y$, we write $d_{TV}(X, Y)$ for the $d_{TV}$ between the underlying distributions of $X$ and $Y$.

We show that given an adaptive / non-adaptive $\mathcal{R}_g$, there is an adaptive / non-adaptive $\mathcal{R}_b$, such that

$$d_{\mathrm{TV}}((R_g^i(s_g^i))_{i\in[T]}, (R_b^i(s_b^i))_{i\in[T]}) = 0. \tag{1}$$

Because the good and bad distributions have identical distributions over the signal space, and this holds for all possible reporting strategies $\mathcal{R}_g$, part (ii) follows immediately.

Consider first non-adaptive reporting. Fix $\mathcal{R}_g = (R_g^1, \ldots, R_g^T)$ where $R_g^i = R_g$ for all $i$, let $\mathcal{R}_b = (R_b^1, \ldots, R_b^T)$, where

$$d_{\mathrm{TV}}(R_g^i(s_g^i), R_b^i(s_b^i)) = 0.$$

The existence of such an $\mathcal{R}_b$ follows from the fact that $d_{\mathrm{DTV}}(g, b) = 0$. Now since $R_g^i(s_g^i)$ and $R_b^i(s_b^i)$ are i.i.d., Equation (1) holds.

Now consider adaptive reporting. For any adaptive reporting strategy $\mathcal{R}_g$, we will construct an adaptive $\mathcal{R}_b$ inductively, such that for any $k$,

$$d_{\mathrm{TV}}((R_g^i(s_g^i))_{i\in[k]}, (R_b^i(s_b^i))_{i\in[k]}) = 0.$$

For the base case when $k = 1$, observe that since $d_{\mathrm{DTV}}(g, b) = 0$, for any $R_g^1$, there exists $R_b^1$ such that

$$d_{\mathrm{TV}}(R_g^1(s_g^1), R_b^1(s_b^1)) = 0.$$

For the inductive case, suppose that $d_{\mathrm{TV}}((R_g^i(s_g^i))_{i\in[k]}, (R_b^i(s_b^i))_{i\in[k]}) = 0$. Given $(R_b^1, \ldots, R_b^k)$, we construct $R_b^{k+1}$ in the following way. Let $R_b^{k+1}$ be such that

$$R_b^{k+1}(s_b^{k+1} \mid \sigma^1, \ldots, \sigma^k) = R_g^{k+1}(s_g^{k+1} \mid \sigma^1, \ldots, \sigma^k),$$

for any $(\sigma^1, \ldots, \sigma^k)$. Now for any $(\sigma^1, \ldots, \sigma^{k+1})$,

$$
\begin{aligned}
&\Pr[(R_b^1(s_b^1), \ldots, R_b^{k+1}(s_b^{k+1})) = (\sigma^1, \ldots, \sigma^{k+1})] \\
&= \Pr[(R_b^1(s_b^1), \ldots, R_b^k(s_b^k)) = (\sigma^1, \ldots, \sigma^k)] \cdot \Pr[R_b^{k+1}(s_b^{k+1} \mid \sigma^1, \ldots, \sigma^k) = \sigma^{k+1}] \\
&= \Pr[(R_g^1(s_g^1), \ldots, R_g^k(s_g^k)) = (\sigma^1, \ldots, \sigma^k)] \cdot \Pr[R_b^{k+1}(s_b^{k+1} \mid \sigma^1, \ldots, \sigma^k) = \sigma^{k+1}] \\
&\hspace{9cm} \text{(induction hypothesis)} \\
&= \Pr[(R_g^1(s_g^1), \ldots, R_g^k(s_g^k)) = (\sigma^1, \ldots, \sigma^k)] \cdot \Pr[R_g^{k+1}(s_g^{k+1} \mid \sigma^1, \ldots, \sigma^k) = \sigma^{k+1}] \\
&\hspace{9cm} \text{(construction of } R_b^{k+1}) \\
&= \Pr[(R_g^1(s_g^1), \ldots, R_g^{k+1}(s_g^{k+1})) = (\sigma^1, \ldots, \sigma^{k+1})].
\end{aligned}
$$

In other words, we have

$$d_{\mathrm{TV}}((R_g^i(s_g^i))_{i\in[k+1]}, (R_b^i(s_b^i))_{i\in[k+1]}) = 0,$$

which concludes the inductive proof for Equation (1) in the adaptive case. $\qquad\square$

*Proof of Theorem 2.* We reduce from Set Cover. More specifically, we use the following decision version of Set Cover: given ground set $X = [n]$, family of sets $\mathcal{F} = \{F_1, \ldots, F_m\}$ where $F_i \subseteq X$, and integer $k = m/2$, determine whether there are $k$ sets in $\mathcal{F}$ whose union is $X$. Note that it is without generality to set $k = m/2$, since given any Set Cover instance with an arbitrary $k$, we could always pad the instance by adding at most $m$ elements into $X$ and $m$ sets into $\mathcal{F}$, to obtain an equivalent new instance with $k' = m'/2$. Fixing a Set Cover instance, we construct $S, \Sigma, E, x$ and $y$ in the following way.

- $S = U \cup V$, where $U = \{u_1, \ldots, u_{n+1}\}$, $V = \{v_1, \ldots, v_{m+2}\}$, and $U \cap V = \varnothing$.
- $\Sigma = \{\sigma_1, \ldots, \sigma_{m+1}\}$.
- $x(u_i) = \frac{1}{2n}$ for $i \in [n]$, and $x(u_{n+1}) = \frac{1}{2}$.
- $y(v_i) = (1/2 + 1/(2n) - t)/m$ for $i \in [m]$, $y(v_{m+1}) = \frac{1}{2} - \frac{1}{2n}$, and $y(v_{m+2}) = t$, where $t \in [0, 1/2 + 1/(2n)]$ is a constant to be determined later.
- For $i \in [m]$ and $j \in F_i$, let $(u_j, \sigma_i) \in E$. Let $(u_{n+1}, \sigma_{m+1}) \in E$.
- For any $i \in [m]$, let $(v_i, \sigma_i) \in E$. For any $i \in [m+2]$, let $(v_i, \sigma_{m+1}) \in E$. For any $i \in [m+1]$, let $(v_{m+1}, \sigma_i) \in E$.
- $E$ contains only edges that mentioned above.

Now consider the problem of finding a set that separates $x$ from $y$ with a positive margin. First observe that such a set $A$ would never include $\sigma_{m+1}$, since $y(\mathrm{pre}(\{\sigma_{m+1}\})) = 1$. Our goal is to set $t$, such that iff $|A| \leq k$ and $\mathrm{pre}(A) = \{u_1, \ldots, u_n\}$, $A$ separates $x$ from $y$ with a positive margin. Such an $A$ in the Set Cover instance would correspond to at most $k$ sets in $\mathcal{F}$ whose union cover $X$. Note that if $\sigma_{m+1} \notin A$,

$$y(\mathrm{pre}(A)) = \frac{1/2 + 1/(2n) - t}{m} \cdot |A| + \frac{1}{2} - \frac{1}{2n} \geq \frac{1}{2} - \frac{1}{2n}.$$

If $\mathrm{pre}(A)$ covers $\{u_1, \ldots, u_n\}$, then $x(\mathrm{pre}(A)) = \frac{1}{2}$. Otherwise, $x(\mathrm{pre}(A)) \leq \frac{1}{2} - \frac{1}{2n} \leq y(\mathrm{pre}(A))$. So if $\mathrm{pre}(A)$ does not cover $\{u_1, \ldots, u_n\}$, $A$ cannot be a separating set. We set $t$ such that $y(\mathrm{pre}(A)) = \frac{1}{2}$ if $|A| = k + 1 = (m+2)/2$. Such a $t$ always exists. Moreover, observe that such a value of $t$ guarantees that whenever $|A| \leq k$, $y(\mathrm{pre}(A)) \leq \frac{1}{2} - \frac{1}{\mathrm{poly}(m,n)}$. Now iff $|A| \leq k$

and $A$ covers $\{u_1, \ldots, u_n\}$, $A$ separates $x$ from $y$ with a margin of $\frac{1}{\text{poly}(m,n)}$. In other words, there is a separating set with a positive margin iff there are at most $k$ sets that cover $X$ in the Set Cover instance. Our NP-hardness result follows. □

*Proof of Theorem 3.* For the first bullet point, let $A^*$ be a set which separates $g$ from $b$ by a margin of $\epsilon$. Consider the following policy: accept $(\sigma^1, \ldots, \sigma^T)$ iff

$$\frac{1}{T} \sum_{i \in [T]} \mathbb{I}[\sigma^i \in A^*] \geq g(\text{pre}(A^*)) - \frac{1}{2}\epsilon.$$

That is, the policy accepts the distribution iff $\bar{\alpha}(A^*) \geq g(\text{pre}(A^*)) - \frac{1}{2}\epsilon$, where $\bar{\alpha}$ is the empirical distribution of the reported signals. We now bound the probability of $g$ being accepted. Using some saturating reporting strategy $R_{A^*}$ for $A^*$ (Proposition 5), we have

$$s_g^i \in \text{pre}(A^*) \iff R_{A^*}(s_g^i) \in A^*.$$

So by the Chernoff-Hoeffding bound, $f$ rejects $g$ with probability

$$\Pr\left[ \frac{1}{T} \sum_i \mathbb{I}[s_g^i \in \text{pre}(A^*)] - g(\text{pre}(A^*)) < -\frac{1}{2}\epsilon \right] \leq \exp(-T\epsilon^2/2) \leq \delta.$$

On the other hand, by Proposition 5 for any reporting strategy $R_b$ of $b$,

$$\Pr[R_b(s_b^i) \in A^*] \leq b(\text{pre}(A^*)) \leq g(\text{pre}(A^*)) - \epsilon.$$

So $f$ accepts $b$ with probability at most

$$\Pr\left[ \frac{1}{T} \sum_i \mathbb{I}[s_b^i \in \text{pre}(A^*)] - b(\text{pre}(A^*)) \geq \frac{1}{2}\epsilon \right] \geq \exp(-T\epsilon^2/2) \leq \delta.$$

For the second bullet point, consider the following instance: $S = \Sigma = (s_1, s_2)$, $g(s_1) = \frac{1}{2} + \epsilon$, $g(s_2) = \frac{1}{2} - \epsilon$, $b(s_1) = b(s_2) = \frac{1}{2}$, and $E = \{(s_1, s_1), (s_2, s_2)\}$. In words, $s_1$ is a good sample/signal, and $s_2$ is a bad one. Agents must report the sample drawn as is. The good distribution draws good samples with slightly higher probability than the bad distribution. For this instance, distinguishing between $g$ and $b$ is exactly equivalent to distinguishing a coin with bias $\epsilon$ with a fair coin. In the latter problem, it is well-known that $\Omega(1/\epsilon^2)$ samples are required. □

*Proof of Theorem 4.* Consider the following policy which uses the policy in Theorem 3 as a building block. Let the policy in Theorem 3 be $f_{g,b}$ for good distribution $g$ and bad distribution $b$. Let $T_0 = 2\ln(k\ell/\delta)/\epsilon^2$, where $\ell T_0 = T$. Given the $T$ reported signals $(\sigma^i)$, our policy $f$ proceeds in the following way:

- For each $i \in [k]$, $j \in [\ell]$, feed the $T_0$ signals

$$\sigma^{(j-1)T_0+1}, \ldots, \sigma^{jT_0}$$

  to policy $f_{g_i, b_j}$, and let the output be $o_{i,j} = f_{g_i, b_j}(\sigma^{(j-1)T_0+1}, \ldots, \sigma^{jT_0})$.
- $f$ outputs 1 iff

$$\bigvee_{i \in [k]} \bigwedge_{j \in [\ell]} o_{i,j} = 1.$$

To see the correctness of the policy, observe that for each any $i, j$, with probability $1 - \frac{\delta}{k\ell}$, $f_{g_i, b_j}$ accepts $g_i$ and rejects $b_j$ given the signals fed in. Taking a union bound over all such $(i, j)$, with probability at least $1 - \delta$, all these policies succeed simultaneously. Now for some good distribution $g_{i^*}$, as long as the above event happens, we have $o_{i^*, j} = 1$ for all $j \in [\ell]$, so

$$\bigvee_{i \in [k]} \bigwedge_{j \in [\ell]} o_{i,j} \geq \prod_{j \in [\ell]} o_{i^*, j} = 1.$$

On the other hand, for some bad distribution $b_{j^*}$, we have $o_{i,j^*} = 0$ for any $i \in [k]$, and therefore

$$\bigvee_{i \in [k]} \bigwedge_{j \in [\ell]} o_{i,j} \leq \sum_i \prod_j o_{i,j} = 0. \qquad \Box$$

Figure 2: Illustration of Proposition 4. Vertices in the frame are from $S$, and the rest of the network is constructed as described in the proof. The dashed edges are saturated in the max flow. The boldface vertices are cut to $s_t$, and therefore constitute the prefix supporting the max separation.

## C   Omitted Proofs From Section 5

*Proof of Proposition 3.* Consider $f'$ (resp. $f''$) which first applies the optimal non-adaptive (resp. adaptive) reporting strategy for $x$ to the original samples, and then applies $f$ to the transformed samples. Now the optimal reporting strategy for $x$ given policy $f'$ (or $f''$) is simply reporting the original sample received from $x$. The proposition follows. □

*Proof of Proposition 4.* We show that $\mathrm{MaxSep}(x, y) = \min_{y \to y'} d_{\mathrm{TV}}(x, y')$, which implies the proposition given Proposition 1.

Consider the following flow network $G = (V, E', w)$:

- $V = S \cup \{s_s, s_t\}$, where $s_s$ is the source and $s_t$ is the sink.
- $E' = E \cup \{(s_s, s)\}_{s \in S} \cup \{(s, s_t)\}_{s \in S}$.
- $w(s_1, s_2) = \infty$ for any $(s_1, s_2) \in E$, $w(s_s, s) = b(s)$ for $s \in S$, and $w(s, s_t) = g(s)$ for $s \in S$.

See Figure 2 for illustration of an example network. Now observe that

- $1 - \mathrm{MaxSep}(x, y)$ is the $s_s$-$s_t$ min-cut of this network. This is because every set $A \subseteq S$ corresponds to a cut, where $S \setminus \mathrm{pre}(A)$ is cut to $s_s$ and $\mathrm{pre}(A)$ is cut to $s_t$. The value of $1 - (x(\mathrm{pre}(A)) - y(\mathrm{pre}(A)))$ is exactly the value of the cut. Similarly, any cut corresponds to a separating set. It follows that $\mathrm{MaxSep}(x, y)$ corresponds to the min-cut.
- $1 - \min_{y \to y'} d_{\mathrm{TV}}(x, y')$ is the $s_s$-$s_t$ max-flow of the network. This is because every $y'$ corresponds to a feasible flow in the network, whose capacity is

$$\sum_s \min\{x(s), y'(s)\} = 1 - d_{\mathrm{TV}}(x, y').$$

Taking $\max$ over $y'$, we see that the max-flow has capacity

$$\max_{y \to y'}(1 - d_{\mathrm{TV}}(x, y')) = 1 - \min_{y \to y'} d_{\mathrm{TV}}(x, y').$$

Strong duality immediately gives the desired statement. □

*Proof of Corollary 1.* Run max-flow on the flow network constructed in the proof of Proposition 4, compute the min-cut on the residual network, and return the subset of $S$ on the same side as $s_s$. □

*Proof of Theorem 5.* Let the policy in Theorem 3 be the *truthful version* of $f_{g,b}$ for good distribution $g$ and bad distribution $b$.[5] Given the $T$ reported signals $(\sigma^i)$, our policy $f$ proceeds in the following way:

521   • For each $i \in [k]$, $j \in [\ell]$, feed all $T$ signals reported to policy $f_{g_i, b_j}$, and let the output be
522     $o_{i,j} = f_{g_i, b_j}(\sigma^1, \dots, \sigma^T)$.
523   • $f$ outputs 1 iff

$$\bigvee_{i \in [k]} \bigwedge_{j \in [\ell]} o_{i,j} = 1.$$

524   The rest of the proof is essentially the same as that of Theorem 4.  □

525   Our policy against any adaptive bad distribution in Theorem 6 uses an efficient learner as a building
526   block, which generalizes classical results for learning discrete distributions.

527   **Theorem 8** (Efficient Learner). *Let $\rho = \rho(G)$ be the width of graph $G = (S, E)$. For any $x \in \Delta(S)$,*
528   *$\epsilon > 0$, $\delta > 0$, and $T = \frac{\rho \ln(1 + n/\rho) \ln(1/\delta)}{2\epsilon^2}$, for any valid reporting strategy that satisfies $(s^i, \sigma^i) \in E$,*
529   *with probability at least $1 - \delta$, $d_{\mathrm{DTV}}(\bar{\alpha}, x) \leq \epsilon$, where $\bar{\alpha}$ is the empirical distribution given by the*
530   *reports $(\sigma^i)_i$, i.e., $\bar{\alpha}(s) = \frac{\sum_i \mathbb{I}[\sigma^i = s]}{T}$.*

531   The following well-known fact about the width is used in the analysis of our learner:

532   **Theorem 9** (Dilworth's Theorem). *A chain in a partially ordered set $G = (S, E)$ is an ordered*
533   *set $C = (c_1, \dots, c_\ell)$, where $c_i \in S$ for $i \in [\ell]$ and $(c_i, c_{i+1}) \in E$ for any $i \in [\ell - 1]$. Dilworth's*
534   *Theorem states that for any partially ordered set $G = (S, E)$, the width of $\rho(G)$ is equal to the*
535   *minimum number of chains whose union covers $S$.*

536   *Proof of Theorem 8.* We show that $\mathrm{MaxSep}(\bar{\alpha}, x) \leq \epsilon$ w.p. $1 - \delta$. More specifically, if for all $A$
537   where $A = \mathrm{pre}(A)$, $\bar{\alpha}(A) - x(A) \leq \epsilon$, then duality gives immediately that $d_{\mathrm{DTV}}(\bar{\alpha}, x) \leq \epsilon$. We
538   will show that this happens with probability $1 - \delta$.

539   Let $\bar{x}$ be the empirical distribution of $(s^i)_i$. Fix $A \subseteq S$ where $A = \mathrm{pre}(A)$. Observe that $\bar{x}(A) \geq$
540   $\bar{\alpha}(A)$, so $x(A) = \mathbb{E}[\bar{x}(A)] \geq \mathbb{E}[\bar{\alpha}(A)]$. The Chernoff bound gives

$$\Pr[\bar{\alpha}(A) \geq x(A) + \epsilon] \leq \exp(-2T\epsilon^2) \leq \frac{\delta}{(1 + n/\rho)^\rho}.$$

541   We only need to show that the number of different sets $A$ where $A = \mathrm{pre}(A)$ is at most $(1 + n/\rho)^\rho$.
542   We call such sets prefixes of graph $(S, E)$. Dilworth's Theorem (Theorem 9) states that the width $\rho$
543   of $(S, E)$ is equal to the minimum number of chains whose union covers $S$. Let $\mathcal{C} = \{C_k\}_{k \in [\rho]}$ be
544   such a covering family, where for any $k$, $C_k = (s_{k,1}, \dots, S_{k,\ell_k})$ is a chain (i.e., $(s_{k,i}, s_{k,i+1}) \in E$
545   for $i \in [\ell_k - 1]$. For any prefix $A$, let $p_k(A) = |A \cap C_k|$. Observe that if two prefixes $A_1$ and $A_2$ are
546   distinct, then there is some $k \in [\rho]$ such that $p_k(A_1) \neq p_k(A_2)$. On the other hand, consider vector
547   $(p_1(A), \dots, p_\rho(A))$. The number of possible values of this vector is $\prod_k (\ell_k + 1) \leq (1 + n/\rho)^\rho$,
548   which is an upper bound of the number of different prefixes. Taking union bound over all these
549   prefixes, we have

$$\Pr[\forall A \text{ where } A = \mathrm{pre}(A), \bar{\alpha}(A) \geq x(A) + \epsilon] \leq \frac{\delta}{(1 + n/\rho)^\rho} \cdot (1 + n/\rho)^\rho = \delta.$$

550   The theorem follows.  □

551   Given the efficient learner constructed above, we are ready to prove Theorem 6.

552   *Proof of Theorem 6.* Consider the following policy: compute the empirical distribution $\bar{\alpha}$ of the
553   reported signals. Accept iff $d_{\mathrm{DTV}}(g, \bar{\alpha}) < \frac{1}{2}\epsilon$. Note that since $g$ is known, $d_{\mathrm{DTV}}(g, \bar{\alpha})$ can be
554   computed in polynomial time using the algorithm in Corollary 1.

555   We first show that $p_{\mathrm{non}}(f, g, T) \geq 1 - \delta$. In particular, we show that if $g$ reports truthfully, then with
556   probability $1 - \delta$, $d_{\mathrm{DTV}}(g, \bar{g}) < 1 - \frac{1}{2}\epsilon$. The argument is similar to that in the proof of Theorem 8.
557   For any $A \subseteq S$ where $A = \mathrm{pre}(A)$, the Chernoff bound implies

$$\Pr[g(A) - \bar{g}(A) \geq \epsilon/2] \leq \frac{\delta}{(1 + n/\rho)^\rho}.$$

558   Since there are at most $(1 + n/\rho)^\rho$ such sets, from a simple union bound, with probability $1 - \delta$,
559   $d_{\mathrm{DTV}}(g, \bar{g}) = \mathrm{MaxSep}(g, \bar{g}) \leq \frac{1}{2}\epsilon$.

560 Now we show that $p_{\mathrm{ada}}(f, b, T) \leq \delta$ for any $b$ where $d_{\mathrm{DTV}}(g, b) \geq \epsilon$. No matter what adaptive
561 reporting strategy $b$ uses, the signals reported by $b$ must satisfy $(s_b^i, \sigma_b^i) \in E$ for all $i$. By Theorem 8,
562 with probability $1 - \delta$, the empirical distribution $\bar{\alpha}$ satisfies $d_{\mathrm{DTV}}(\bar{\alpha}, b) \leq \frac{1}{2}\epsilon$. Now since $d_{\mathrm{DTV}}$
563 satisfies the triangle inequality (Proposition 2),

$$d_{\mathrm{DTV}}(g, \bar{\alpha}) \geq d_{\mathrm{DTV}}(g, b) - d_{\mathrm{DTV}}(\bar{\alpha}, b) \geq \epsilon - \frac{1}{2}\epsilon = \frac{1}{2}\epsilon.$$

564 Whenever this happens, $b$ is rejected by $f$, which means $p_{\mathrm{ada}}(f, b, T) \leq \delta$. $\qquad\square$

565 *Proof of Theorem 7.* We use the algorithm by Valiant and Valiant [17] for testing identity of discrete
566 distributions as a building block. Given a distribution $x \in \Delta([n])$, with $T = O\left(\frac{\sqrt{n} \ln(1/\delta)}{\epsilon^2}\right)$ samples
567 to an unknown distribution $y$, their algorithm distinguishes between the following two cases: (1)
568 $y = x$ and (2) $d_{\mathrm{TV}}(x, y) \geq \epsilon$. Our policy for non-adaptive reporting is simply running the algorithm
569 by Valiant and Valiant on the good distribution $g$ and the signals reported $(\sigma^i)_i$.

570 The good distribution $g$, in order to be accepted with high probability, simply reports truthfully. The
571 distribution of signals of $g$ is therefore exactly $g$, which with probability $1 - \delta$ passes the test.

572 As for the bad distribution, observe that any non-adaptive reporting strategy $\mathcal{R}_b = (R_b, \ldots, R_b)$
573 induces a distribution $\alpha_b$ of signals reported, where $b \rightarrow_{R_b} \alpha_b$. No matter how $b$ reports, because
574 $d_{\mathrm{DTV}}(g, b) \geq \epsilon$, we always have $d_{\mathrm{TV}}(g, \alpha_b) \geq \epsilon$, in which case $\alpha_b$ fails the test with probability at
575 least $1 - \delta$. $\qquad\square$