[Reviews · NeurIPS 2019]

Reviewer 1



The paper introduces a new model of strategic classification. Given some type-dependent feature distribution, agents can transform these features into "signals" according to some background graph. The principal then classifies agents based on the signals. The model is a departure from many recent models of strategic classification, which frame agents as maximizing utility subject to a cost function penalty, and also deals with the setting of repeated samples from agents rather than a one-shot game. This has the benefit of elucidating the importance of differences in the agents' initial feature distribution (in terms of DTV) that may be intuitively true, but has not been captured in recent work. On the other hand, the paper is missing a more thorough comparison between these different perspectives. I find the perspective of manipulating features more natural for many strategic classification problems, and the machine learning paper example in the paper is pretty toy. Are there more natural settings when adopting this graph perspective makes sense? In these cases, where does the background graph come from? Within this graph-theoretic model, the paper presents a complete structural results for when successful classification is/is not possible in terms of DTV for general graphs, as well as a hardness result for testing whether DTV is non-zero, and so above chance classification is possible, or not. The sample-complexity result is less exciting and follows almost immediately from Hoeffding's inequality once the definitions are in place. It also assumes access to a separating set, which the Theorem 2 shows is NP-Hard to compute in general. Given the general hardness results, I appreciated Section 5, which deals with "partially ordered" signals and shows, for some graphs, efficient algorithms are possible. However, the motivation for "partially ordered" signals is weak and de-emphasizes the role of graph structure in determining whether inference is possible. I would have appreciated more discussion on what types of graph structures make classification possible in principle--- is there a more general characterization? Overall, the paper is fairly clear and well-written, though I found the introduction too heavy-handed. I am not entirely convinced by the utility of the modeling formalism advocated in this paper for strategic classification problems. Within this formalism, the structural results neatly characterize when inference is possible in general, and the NP-Hardness results stress the need to consider graphs with more structure. The example of partially ordered signal sets nicely shows the benefits of structure, though a broader characterization of what types of structures make classification possible would significantly strengthen this paper. After rebuttal: Thanks to the authors for their thoughtful response. I still think the paper would benefit from a clearer comparison between modeling approaches to strategic classification. I also think the commentary on the revelation principle provided in the rebuttal should be expanded in Section 5. I am, however, raising my score because I was persuaded of the potential applicability of the model through the given examples, and I appreciate the structural characterization provided in terms of new quantities like DTV and MaxSep that allows you to clearly reason about this model.

Reviewer 2



The authors propose a setting where agents receive a sample (e.g. a research idea), and they can transform samples into one of multiple signals (e.g. a conference). The ability to convert certain samples into signals is given by a bipartite graph. There are two types of agents, “good” and “bad”, which both try to appear as being good, while the goal for the paper is to come up with an approach (requiring as few samples as possible) to differentiate between the two. The authors show that the sample complexity to differentiate between good and bad is mild and dependent on the directed total variational distance (d_DTV) between the good and bad distributions. On the other hand, it’s NP-hard to distinguish if the d_DTV is close to, or identical to, 0. Finally, the authors show that in a special case where signals are partially ordered, the sample complexity goes down and the hardness result disappears. The paper proposed a nice model to study learning from strategically manipulated samples. The results in this model are exciting, and point at interesting potential follow-ups. After author response: Since I had raised no serious concerns, the authors feedback did not need to address my comments, hence my review and score remains the same.

Reviewer 3



This is a solid paper. The model studied in the paper is well-motivated and has many applications. I found the example of hiring ML researchers in the introduction very helpful. It provides a clear description and motivation of the model. I like the equivalence between DTV and MaxSep. In particular, it implies that the principal’s optimal policy has a simple form: just binary classify the signals into “good” and “bad” then count the number of “good” signals to decide whether the distribution is good. Even though the principal can use much more complex policies, it is surprising that the optimal policy has such an intuitive form. TYPOS Line 477 in Page 13: ">= exp(-T*epsilon^2/2)" should be "<=".

[Author Response · NeurIPS 2019]

Thank you for your detailed and helpful reviews! Reviewers 2 and 3 are positive about the paper and don't have specific questions, so we will focus on responding to Reviewer 1.

Reviewer 1's main concern seems to be the applicability of the model, pointing out that the example in the introduction is "toy," and being concerned about the NP-hardness of the problem in the fully general case, saying she/he appreciates Section 5 where everything can be done efficiently, but wondering whether we have done enough in terms of characterizing easy cases. Let us address these concerns next.

First, the purpose of the example in the introduction is more to illustrate the model (and Reviewer 3 liked the example for that reason) than to illustrate a direct practical application. The most obvious applications are in the special case where the agent has some data and can withhold some of these data, or even parts of individual examples. (Indeed, this is the case that motivated us to study this topic in the first place; but in the end, we had results for a much more general model so we decided to present it accordingly.)

- For example, we can think about someone applying to a music school and submitting recordings of her performances — but she could easily omit some recordings, or cut parts of recordings out. There are many similar examples involving not musicians but athletes, actors, etc.

- Another example that we took out of the paper due to space constraint is a publisher who is trying to convince an advertiser that high-value users visit her site during a trial period. In this case, the publisher can decide to not show low-value users the ad, or direct them to a different part of the site.

These examples illustrate the kinds of settings for which our graph-theoretic framework makes more sense than the costly feature manipulation framework. E.g., conceptually, for the example of cutting off the end and beginning of a recording, the graph would have edges from intervals (of the recording) to all subintervals. For the example where the publisher shields the advertiser from certain users, the graph would have edges from every user (sample) to both itself and the "empty" sample/signal (no user is shown).

These examples, and indeed all the settings that originally motivated this work, also exactly fit the special case in Section 5, so everything is tractable too. Section 5 captures a very general class of applications, which still involves a misreporting graph. While it is true that there may be other special cases that are also tractable, in general, there is no good way to characterize all special cases that are not NP-hard. In terms of economic theory, there is a very natural distinction between Section 5, where the key aspect is that the revelation principle holds (i.e., without loss of generality, good types can report truthfully) and the more general case where the revelation principle does not hold. (In other, non-learning contexts in mechanism design, whether the revelation principle holds is often an aspect that determines whether things are tractable.)

We will add these examples to the paper; we agree that they should be helpful to the reader.

We thank all the reviewers again for their detailed and encouraging comments!

[Meta-Review · NeurIPS 2019]

Reviewers recommend acceptance based on the model presented, which is interesting and useful, and the structural results provided. While there was consensus that the paper was clearly written, please take note of the constructive feedback provided, such as the comparison to strategic classification and discussion of the revelation principle.